# HIERARCHICAL VISUOMOTOR CONTROL OF HUMANOIDS

**Josh Merel**[*] **Arun Ahuja**[*]
**Vu Pham, Saran Tunyasuvunakool, Siqi Liu, Dhruva Tirumala,**
**Nicolas Heess & Greg Wayne**
DeepMind
London, UK
`{jsmerel,arahuja,vuph,stunya,liusiqi,dhruvat,`
` heess,gregwayne}@google.com`

## ABSTRACT

We aim to build complex humanoid agents that integrate perception, motor control, and memory. In this work, we partly factor this problem into low-level motor control from proprioception and high-level coordination of the low-level skills informed by vision. We develop an architecture capable of surprisingly flexible, task-directed motor control of a relatively high-DoF humanoid body by combining pre-training of low-level motor controllers with a high-level, task-focused controller that switches among low-level sub-policies. The resulting system is able to control a physically-simulated humanoid body to solve tasks that require coupling visual perception from an unstabilized egocentric RGB camera during locomotion in the environment. Supplementary video link[1]

## 1 INTRODUCTION

In reinforcement learning (RL), a major challenge is to simultaneously cope with high-dimensional input and high-dimensional action spaces. As techniques have matured, it is now possible to train high-dimensional vision-based policies from scratch to generate a range of interesting behaviors ranging from game-playing to navigation (Jaderberg et al., 2018; OpenAI, 2018; Wayne et al., 2018). Likewise, for controlling bodies with a large number of degrees of freedom (DoFs), in simulation, reinforcement learning methods are beginning to surpass optimal control techniques. Here, we try to synthesize this progress and tackle high-dimensional input and output at the same time. We evaluate the feasibility of full-body *visuomotor* control by comparing several strategies for humanoid control from vision.

Both to simplify the engineering of a visuomotor system and to reduce the complexity of task-directed exploration, we construct modular agents in which a high-level system possessing egocentric vision and memory is coupled to a low-level, reactive motor control system. We build on recent advances in imitation learning to make flexible low-level motor controllers for high-DoF humanoids. The motor skills embodied by the low-level controllers are coordinated and sequenced by the high-level system, which is trained to maximize sparse task reward.

Our approach is inspired by themes from neuroscience as well as ideas developed and made concrete algorithmically in the animation and robotics literatures. In motor neuroscience, studies of spinal reflexes in animals ranging from frogs to cats have led to the view that locomotion and reaching are highly prestructured, enabling subcortical structures such as the basal ganglia to coordinate a motor repertoire; and cortical systems with access to visual input can send low complexity signals to motor systems in order to evoke elaborate movements (Flash & Hochner, 2005; Bizzi et al., 2008; Grillner et al., 2005).

The study of "movement primitives" for robotics descends from the work of Ijspeert et al. (2002). Subsequent research has focused on innovations for learning or constructing primitives for control

---

[*]Equal contribution.

[1]`https://youtu.be/7GISvfbykLE`

of movments (Ijspeert et al., 2003; Kober & Peters, 2009), deploying and sequencing them to solve tasks (Sentis & Khatib, 2005; Kober & Peters, 2014; Konidaris et al., 2012), and increasing the complexity of the control inputs to the primitives (Neumann et al., 2014). Particularly relevant to our cause is the work of Kober et al. (2008) in which primitives were coupled by reinforcement learning to external perceptual inputs.

Research in the animation literature has also sought to produce physically simulated characters capable of distinct movements that can be flexibly sequenced. This ambition can be traced to the virtual stuntman (Faloutsos et al., 2001b;a) and has been advanced markedly in the work of Liu (Liu et al., 2012). Further recent work has relied on reinforcement learning to schedule control policies known as "control fragments", each one able to carry out only a specialized short movement segment (Liu & Hodgins, 2017; 2018). In work to date, such control fragments have yet to be coupled to visual input as we will pursue here. From the perspective of the RL literature (Sutton et al., 1999), motor primitives and control fragments may be considered specialized instantiations of "option" sub-policies.

Our work aims to contribute to this multi-disciplinary literature by demonstrating concretely how control-fragment-like low-level movements can be coupled to and controlled by a vision and memory-based high-level controller to solve tasks. Furthermore, we demonstrate the scalability of the approach to greater number of control fragments than previous works. Taken together, we demonstrate progress towards the goal of integrated agents with vision, memory, and motor control.

## 2 APPROACH

We present a system capable of solving tasks from vision by switching among low-level motor controllers for the humanoid body. This scheme involves a general separation of control where a low-level controller handles motor coordination and a high-level controller signals/selects low-level behavior based on task context (see also Heess et al. 2016; Peng et al. 2017). In the present work, the low-level motor controllers operate using proprioceptive observations, and the high-level controller operate using proprioception along with first-person/egocentric vision. We first describe the procedure for creating low-level controllers from motion capture data, then describe and contrast multiple approaches for interfacing the high- and low-level controllers.

### 2.1 TRACKING MOTION CAPTURE CLIPS

For simulated character control, there has been a line of research extracting humanoid behavior from motion capture ("mocap") data. The SAMCON algorithm is a forward sampling approach that converts a possibly noisy, *kinematic* pose sequence into a physical trajectory. It relies on a beam-search-like planning algorithm (Liu et al., 2010; 2015) that infers an action sequence corresponding to the pose sequence. In subsequent work, these behaviors have been adapted into policies (Liu et al., 2012; Ding et al., 2015). More recently, RL has also been used to produce time-indexed policies which serve as robust tracking controllers (Peng et al., 2018). While the resulting time-indexed policies are somewhat less general as a result, time-indexing or phase-variables are common in the animation literature and also employed in kinematic control of characters (Holden et al., 2017). We likewise use mocap trajectories as reference data, from which we derive policies that are single purpose – that is, each policy robustly tracks a short motion capture reference motion (2-6 sec), but that is all each policy is capable of.

**Humanoid body** We use a 56 degree-of-freedom (DoF) humanoid body that was developed in previous work (Merel et al., 2017), a version of which is available with motion-capture playback in the DeepMind control suite (Tassa et al., 2018). Here, we actuate the joints with position-control: each joint is given an actuation range in $[-1, 1]$, and this is mapped to the angular range of that joint.

**Single-clip tracking policies** For each clip, we train a policy $\pi_\theta(a|s, t)$ with parameters $\theta$ such that it maximizes a discounted sum of rewards, $r_t$, where the reward at each step comes from a custom scoring function (see eqns. 1, 2 defined immediately below). This tracking approach most closely follows Peng et al. (2018). Note that here the state optionally includes a normalized time $t$ that goes from 0 at the beginning of the clip to 1 at the end of the clip. For cyclical behaviors like locomotion, a gait cycle can be isolated manually and kinematically blended circularly by weighted

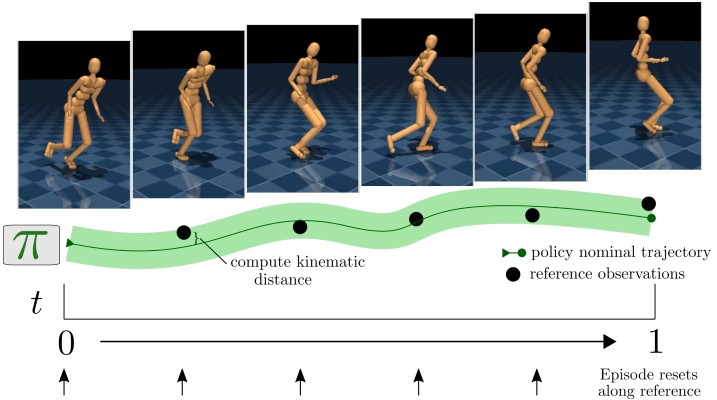

Figure 1: Illustration of tracking-based RL training. Training iteratively refines a policy to robustly track the reference trajectory as well as physically feasible.

linear interpolation of the poses to produce a repeating walk. The time input is reset each gait-cycle (i.e. it follows a sawtooth function). As proposed in Merel et al. (2017); Peng et al. (2018), episodes are initialized along the motion capture trajectory; and episodes can be terminated when it is determined that the behavior has failed significantly or irrecoverably. Our specific termination condition triggers if parts of the body other than hands or feet make contact with the ground. See Fig. 1 for a schematic.

We first define an energy function most similar to SAMCON's (Liu et al., 2010):

$$E_{total} = w_{qpos}E_{qpos} + w_{qvel}E_{qvel} + w_{ori}E_{ori}+ \\ w_{ee}E_{ee} + w_{vel}E_{vel} + w_{gyro}E_{gyro} \tag{1}$$

where $E_{qpos}$ is a energy defined on all joint angles, $E_{qvel}$ on joint velocities, $E_{ori}$ on the body root (global-space) quaternion, $E_{ee}$ on egocentric vectors between the root and the end-effectors (see Merel et al. (2017)), $E_{vel}$ on the (global-space) translational velocities, and $E_{gyro}$ on the body root rotational velocities. More specifically:

$$E_{qpos} = \frac{1}{N_{qpos}} \sum |\vec{q}_{pos} - \vec{q}_{pos}^\star| \qquad\qquad E_{ee} = \frac{1}{N_{ee}} \sum ||\vec{q}_{ee} - \vec{q}_{ee}^\star||_2$$

$$E_{qvel} = \frac{1}{N_{qvel}} \sum |\vec{q}_{vel} - \vec{q}_{vel}^\star| \qquad\qquad E_{vel} = 0.1 \cdot \frac{1}{N_{vel}} \sum |\vec{x}_{vel} - \vec{x}_{vel}^\star|$$

$$E_{ori} = || \log(\vec{q}_{ori} \cdot \vec{q}_{ori}^{\star-1})||_2 \qquad\qquad E_{gyro} = 0.1 \cdot ||\vec{q}_{gyro} - \vec{q}_{gyro}^\star||_2$$

where $\vec{q}$ represents the pose and $\vec{q}^\star$ represents the reference pose. In this work, we used coefficients $w_{qpos} = 5$, $w_{qvel} = 1$, $w_{ori} = 20$, $w_{gyro} = 1$, $w_{vel} = 1$, $w_{ee} = 2$. We tuned these by sweeping over parameters in a custom implementation of SAMCON (not detailed here), and we have found these coefficients tend to work fairly well across a wide range of movements for this body.

From the energy, we write the reward function:

$$r_t = \exp(-\beta E_{total}/w_{total}) \tag{2}$$

where $w_{total}$ is the sum of the per energy-term weights and $\beta$ is a sharpness parameter ($\beta = 10$ throughout). Since all terms in the energy are non-negative, the reward is normalized $r_t \in (0, 1]$ with perfect tracking giving a reward of 1 and large deviations tending toward 0.

Acquiring reference data features for some quantities required setting the body to the pose specified by the joint angles: e.g., setting $\vec{x}_{pos}$, $\vec{q}_{pos}$, and $\vec{q}_{ori}$ to compute the end-effector vectors $\vec{q}_{ee}$. Joint angle velocities, root rotational velocities, and translational velocities ($\vec{q}_{vel}$, $\vec{q}_{gyro}$, $\vec{x}_{vel}$) were derived from the motion capture data by finite difference calculations on the corresponding positions. Note that the reward function here was not restricted to egocentric features – indeed, the velocity and quaternion were non-egocentric. Importantly, however, the policy received exclusively egocentric observations, so that, for example, rotating the initial pose of the humanoid would not affect the policy's ability to execute the behavior. The full set of proprioceptive features we provided the policy consists of joint angles ($\vec{q}_{pos}$) and velocities ($\vec{q}_{vel}$), root-to-end-effector vectors ($\vec{q}_{ee}$), root-frame velocimeter ($\vec{q}_{veloc}$), rotational velocity ($\vec{q}_{gyro}$), root-frame accelerometers ($\vec{q}_{accel}$), and 3D orientation relative to the z-axis ($\vec{r}_z$: functionally a gravity sensor).

**Low-level controller reinforcement learning details**    Because the body is position-controlled, ($a_t$ has the same dimension and semantics as a subset of the body pose), we can pre-train the policy to produce target poses by supervised learning $\max_\theta \sum_t \log \pi(q^*_{pos,t+1}|s^*_t, t)$. This produces very poor control but facilitates the subsequent stage of RL-based imitation learning. We generally found that training with some pretraining considerably shortened the time the training took to converge and improved the resulting policies.

For RL, we performed off-policy training using a distributed actor-critic implementation, closest to that used in (Hausman et al., 2018). This implementation used a replay buffer and target networks as done in previous work (Lillicrap et al., 2015; Heess et al., 2015). The Q-function was learned off-policy using TD-learning using importance-weighted Retrace (Munos et al., 2016), and the actor was learned off-policy using SVG(0) (Heess et al., 2015). This is to say that we learned the policy by taking gradients with respect to the Q function (target networks were updated every 500 learning steps). Gradient updates to the policy were performed using short time windows, $\{s_\tau, a_\tau\}_{\tau=1...T}$, sampled from replay:

$$\max_{\pi_\theta} \sum_{\tau=1...T} \mathbb{E}_{a \sim \pi(a|s_\tau)}[Q_{target}(s_\tau, a)] - \eta \mathcal{D}_{KL}[\pi_\theta(a|s_\tau)||\pi_{target}(a|s_\tau)] \qquad (3)$$

where $\eta$ was fixed in our experiments. While the general details of the RL algorithm are not pertinent to the success of this approach (e.g. Peng et al. (2018) used on-policy RL), we found two details to be critical, and both were consistent with the results reported in Peng et al. (2018). Policy updates needed to be performed conservatively with the update including a term which restricts $\mathcal{D}_{KL}[\pi_{new}||\pi_{old}]$ (Heess et al., 2015; Schulman et al., 2017). Secondly, we found that attempting to learn the variance of the policy actions tended to result in premature convergence, so best results were obtained using a stochastic policy with fixed noise (we used noise with $\sigma = .1$).

## 2.2    Varieties of low-level motor control

We next consider how to design low-level motor controllers derived from motion capture trajectories. Broadly, existing approaches fall into two categories: *structured* and *cold-switching* controllers. In structured controllers, there is a hand-designed relationship between "skill-selection" variables and the generated behavior. Recent work by Peng et al. (2018) explored specific hand-designed, structured controllers. While parameterized skill-selection coupled with manual curation and preprocessing of motion capture data can produce artistically satisfying results, the range of behavior has been limited and implementation requires considerable expertise and animation skill. By contrast, an approach in which behaviors are combined by a more automatic procedure promises to ultimately scale to a wider range of behaviors.

Below, we describe some specific choices for both structured and cold-switching controllers. For structured control schemes, we consider: (1) a *steerable* controller that produces running behavior with a controllable turning radius, and (2) a *switching* controller that is a single policy that can switch between the behaviors learned from multiple mocap clips, with switch points allowed at the end of gait cycles. The allowed transitions were defined by a transition graph. For cold switching, we will not explicitly train transitions between behaviors.

**Steerable controller**    Following up on the ability to track a single cyclical behavior like locomotion described above, we can introduce the ability to parametrically turn. To do this we distorted the reference trajectory accordingly and trained the policy to track the reference with the turning radius as additional input. Each gait cycle we picked a random turning radius parameter and in that gaitcyle we rotate the reference clip heading ($\vec{q}_{ori}$) at that constant rate (with appropriate bookkeeping for other positions and velocities). The result was a policy that, using only one gait cycle clip as input, could turn with a specified rate of turning.

**Switching controller**    An alternative to a single behavior with a single continuously controllable parameter is a single policy that is capable of switching among a discrete set of behaviors based on a 1-of-$k$ input. Training consisted of randomly starting in a pose sampled from a random mocap clip and transitioning among clips according to a graph of permitted transitions. Given a small, discrete set of clips that were manually "cut" to begin and end at similar points in a gait cycle, we initialized a discrete Markov process among clips with some initial distribution over clips and transitioned between clips that were compatible (walk forward to turn left, etc.) (Fig. 2).

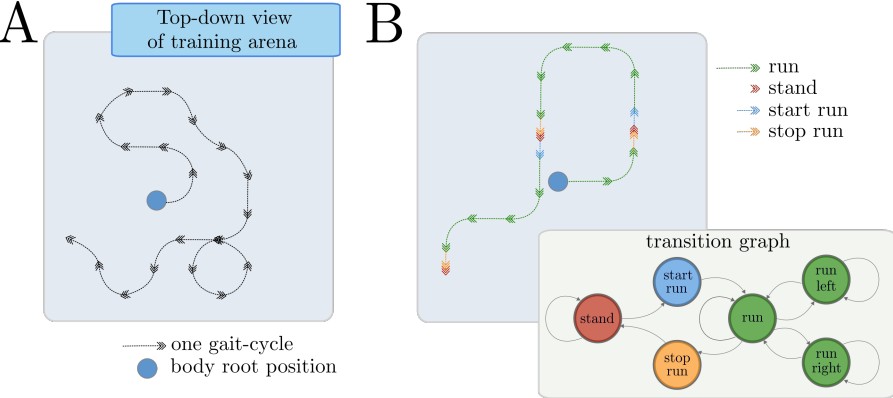

Figure 2: Training settings for explicit training of transition-capable controllers. Panel A depicts a cartoon of a training episode for a steerable controller in which the turning radius of a each gait-cycle is selected randomly. Panel B depicts training a policy under an explicit, hand-designed transition graph for $k$ options.

**Cold-switching of behaviors and control fragments** We can also leave the task of sequencing behaviors to the high-level controller, instead of building structured low-level policies with explicit, designed transitions. Here, we did not attempt to combine the clips into a single policy; instead, we cut behaviors into short micro-behaviors of roughly 0.1 to 0.3 seconds, which we refer to as *control fragments* (Liu & Hodgins, 2017). Compared to switching using the complete behaviors, the micro-behaviors, or control fragments, allow for better transitions and more flexible locomotion. Additionally, we can easily scale to many clips without manual intervention. For example, clip 1 would generate a list of fragments: $\pi_1^1(a|s_t, \tau), \pi_2^1(a|s_t, \tau), \ldots, \pi_{10}^1(a|s_t, \tau)$. When fragment 1 was chosen, $\tau$ the time-indexing variable was set to $\tau = 0$ initially and ticked until, say, $\tau = 0.1$. Choosing fragment 2, $\pi_2^1$, would likewise send a signal to the clip 1 policy starting from $\tau = 0.1$, etc. Whereas we have to specify a small set of consistent behaviors for the other low-level controller models, we could easily construct hundreds (or possibly more) control fragments cheaply and without significant curatorial attention. Since the control fragments were not trained with switching behavior, we refer to the random access switching among fragments by the high-level controller as "cold-switching" (Fig. 3).

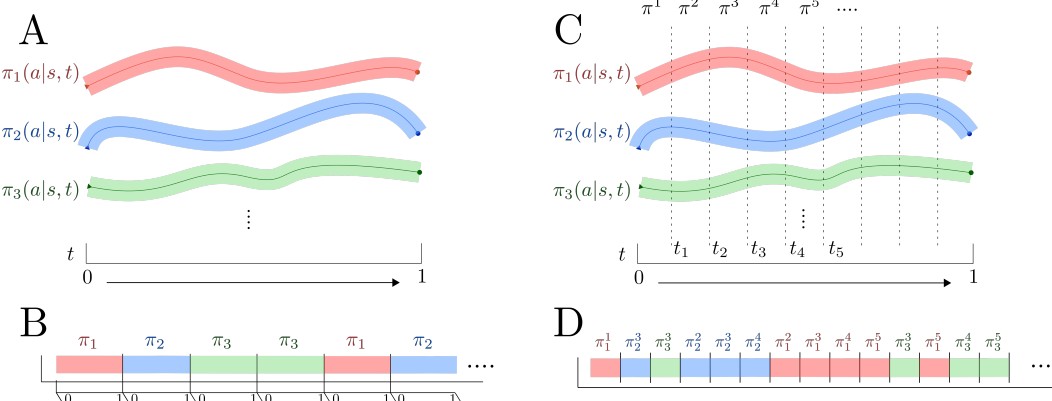

Figure 3: Cold-switching among a set of behaviors (A) only at end of clips to form a trajectory composed of sequentially activation of the policies (B). Alternatively, policies are fragmented at a pre-specified set of times, cutting the policy into sub-policies (C), which serve as *control fragments*, enabling sequencing at a higher frequency (D).

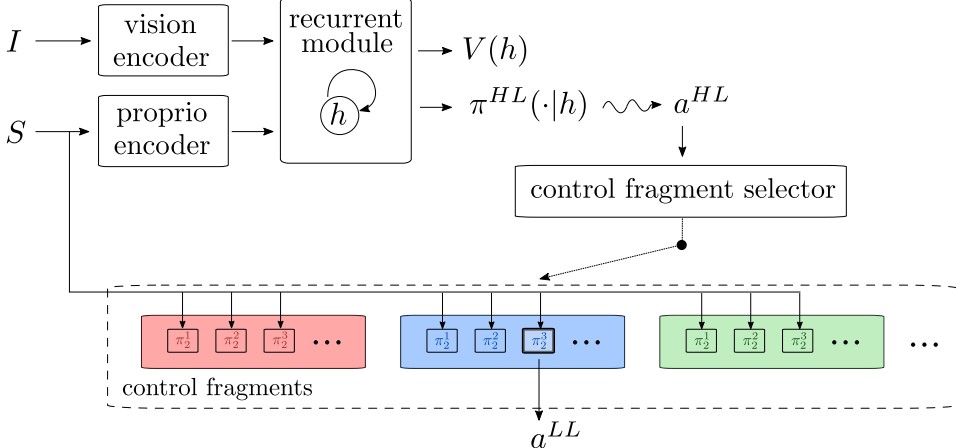

Figure 4: Schematic of the architecture: a high-level controller (HL) selects among multiple low-level (LL) control fragments, which are policies with proprioception. Switching from one control fragment to another occurs every $k$ time steps.

## 2.3 TRAINING HL-POLICIES TO SOLVE TASKS USING LL-CONTROLLERS

We integrated the low-level controllers into an agent architecture with vision and and an LSTM memory in order to apply it to tasks including directed movements to target locations, a running course with wall or gap obstacles, a foraging task for "balls", and a simple memory task involving detecting and memorizing the reward value of the balls.

The interface between the high-level controller and the low-level depends on the type of low-level controller: for the *steerable* controller, the high-level produces a one-dimensional output; for the *switching* and *control fragment* controllers, the high-level produces a 1-of-K index to select the low-level policies. The high-level policies are trained off-policy using data from a replay buffer. The replay buffer contains data generated from distributed actors, and in general the learner processes the same replay data multiple times.

The high-level controller senses inputs from proprioceptive data and, for visual tasks, an egocentric camera mounted at the root of the body (Fig. 4). A noteworthy challenge arises due to the movement of the camera itself during locomotion. The proprioceptive inputs are encoded by a single linear layer, and the image is encoded by a ResNet (see Appendix A). The separate inputs streams are then flattened, concatenated, and passed to an LSTM, enabling temporally integrated decisions, with a stochastic policy and a value function head. The high-level controller receives inputs at each time step even though it may only act when the previous behavior (gait cycle or control fragment) has terminated.

Importantly, while the low-level skills used exclusively egocentric proprioceptive input, the high-level controller used vision to select from or modulate them, enabling the system as a whole to effect visuomotor computations.

**High-level controller reinforcement learning details** For the *steerable* controller, the policy was a parameterized Gaussian distribution that produces the steering angle $a_s \in [-1.5, 1.5]$. The mean of Gaussian was constrained via a $\tanh$ and sampled actions were clipped to the steering angle range. The steering angle was held constant for a full gait cycle. The policy was trained as previously described by learning a state-action value function off-policy using TD-learning with Retrace (Munos et al., 2016) with the policy trained using SVG(0) (Heess et al., 2015).

For the *switching* controller and the discrete *control fragments* approach, the policy was a multinomial over the discrete set of behaviors. In either case, the high-level controller would trigger the behavior for its period $T$ (a gait cycle or a fragment length). To train these discrete controllers, we fit the state-value baseline $V$-function using V-Trace and update the policy according to the method in Espeholt et al. (2018). While we provided a target for the value function loss at each time step,

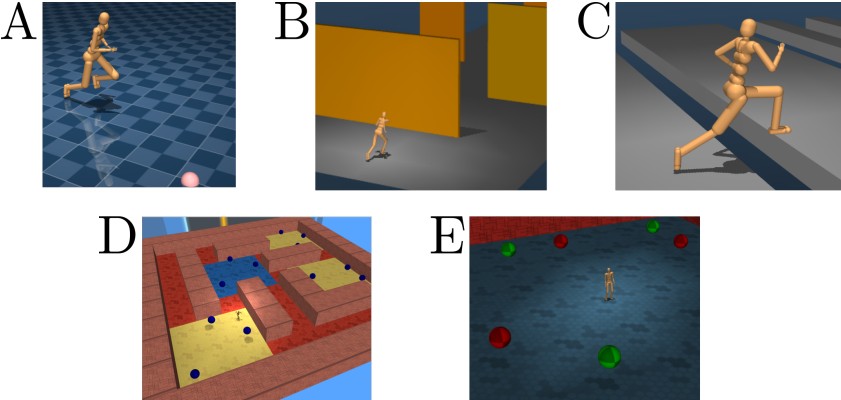

Figure 5: A. Go-to-target: in this task, the agent moves on an open plane to a target provided in egocentric coordinates. B. Walls: The agent runs forward while avoiding solid walls using vision. C. Gaps: The agent runs forward and must jump between platforms to advance. D. Forage: Using vision, the agent roams in a procedurally-generated maze to collect balls, which provide sparse rewards. E. Heterogeneous Forage: The agent must probe and remember rewards that are randomly assigned to the balls in each episode.

the policy gradient loss for the high-level was non-zero only when a new action was sampled (every $T$ steps).

**Query-based control fragment selection**   We considered an alternative family of ideas to interface with control fragments based on producing a Gaussian policy search query to be compared against a feature-key for each control fragment. We then selected the control fragment whose key was nearest the query-action. Our method was based on the Wolpertinger approach introduced in (Dulac-Arnold et al., 2015). Here, the Q-function was evaluated for each of $k$ nearest neighbors to the query-action, and the control fragment were selected with Boltzmann exploration, i.e. $p(a_i^{HL}|h) \propto \exp(\frac{1}{T}Q(h, a_i^{HL}))$, where $h$ is the output of the LSTM. See Appendix A.3.3 for more details. The intuition was that this would allow the high-level policy to be less precise as the Q-function could assist it in selecting good actions. However, this approach under-performed relative to discrete action selection as we show in our results.

## 3   Experiments

### 3.1   Results on core tasks

We compared the various approaches on a variety of tasks implemented in MuJoCo (Todorov et al., 2012). The core tasks we considered for the main comparisons were Go-to-target, wall navigation (Walls), running on gapped platforms (Gaps), foraging for colored ball rewards (Forage), and a foraging task requiring the agent to remember the reward value of the different colored balls (Heterogeneous Forage) (see Fig. 5). In Go-to-target, the agent received a sparse reward of 1 for each time step it was within a proximity radius of the target. For Walls and Gaps, adapted from Heess et al. (2017) to operate from vision, the agent received a reward proportional to its forward velocity. Forage was broadly similar to *explore_object_locations* in the DeepMind Lab task suite (Beattie et al., 2016) (with a humanoid body) while Heterogeneous Forage was a simplified version of *explore_object_rewards*. In all tasks, the body was initialized to a random pose from a subset of the reference motion capture data. For all tasks, other than Go-to-target, the high-level agent received a 64x64 image from the camera attached to the root of the body, in addition to the proprioceptive information.

We compared the agents on our core set of tasks. Our overall best results were achieved using control fragments with discrete selection (Fig. 6). Additional training details are provided in Appendix A. For comparison, we also include the control experiment of training a policy to control the humanoid from scratch (without low-level controllers) as well as training a simple rolling ball body. The

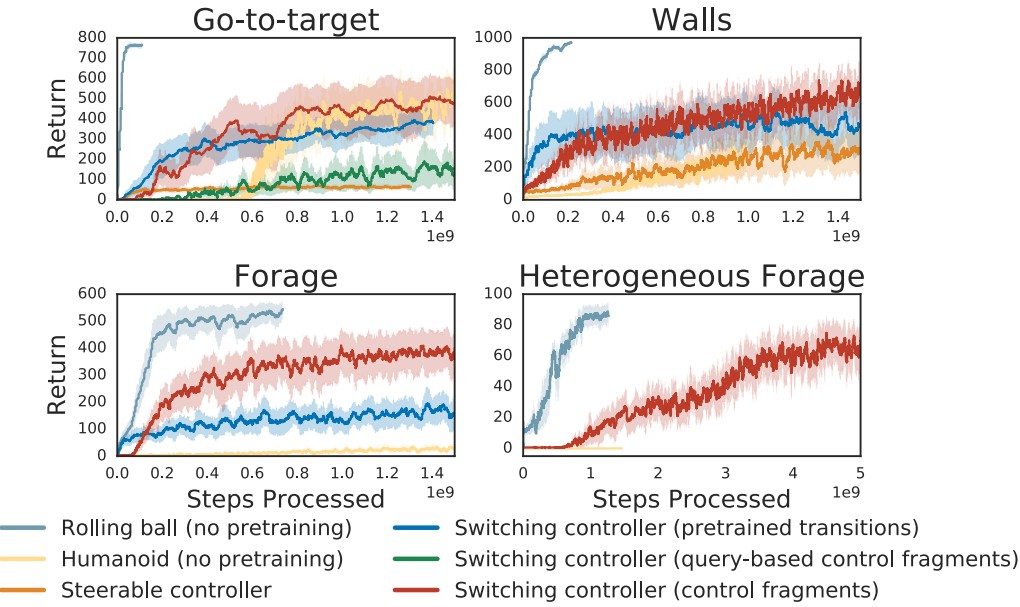

Figure 6: Performance of various approaches on each core task. Of the approaches we compared, discrete switching among control fragments performed the best. Plots show the mean and standard error over multiple runs.

performance of the rolling ball is not directly comparable because its velocity differs from that of the humanoid, but isolates the task complexity from the challenge of motor control of the humanoid body. The switching controllers selected between a base set of four policies: stand, run, left and right turn. For the control fragments approach we were able to augment this set as described in Table 2.

The end-to-end approach (described in Appendix A.4) succeeded at only Go-to-target, however the resulting visual appearance was jarring. In the more complex Forage task, the end-to-end approach failed entirely. The steering controller was also able to perform the Go-to-target task, but a fixed turning radius meant that it was unable to make a direct approach the target, resulting in a long travel time to the target and lower score. Both the steering controller and switching controller were able to reach the end of the course in the Walls task, but only the control fragments approach allowed for sharper turns and quicker adjustments for agent to achieve a higher velocity. Generally, the switching controller with transitions started to learn faster and appeared the most graceful because of its predefined, smooth transitions, but its comparative lack of flexibility meant that its asymptotic task performance was relatively low. In the Forage task, where a score of $> 150$ means the agent is able to move around the maze and 600 is maximum collection of reward, the switching controller with transitions was able to traverse the maze but unable to adjust to the layout of the maze to make sharper turns to collect all objects. The control fragments approach was able to construct rotations and abrupt turns to collect the objects in each room. In the Gaps task, we were able to use the control fragments approach with 12 single-clip policies, where it would be laborious to pretrain transitions for each of these. In this task, the high-level controller selected between the 4 original stand, run and turn policies as well as 8 additional jumps, resulting in 359 fragments, and was able to synthesize them to move forward along the separated platforms. In the final Heterogeneous Forage task, we confirmed that the agent, equipped with an LSTM in the high-level controller, was capable of memory-dependent control behavior. See our Extended Video[2] for a comprehensive presentation of the controllers.

All control fragment comparisons above used control fragments of 3 time steps (0.09s). To further understand the performance of the control fragment approach, we did a more exhaustive comparison of performance on Go-to-target of the effect of fragment length, number of fragments, as well as

---

[2]https://youtu.be/dKM--__Q8NQ

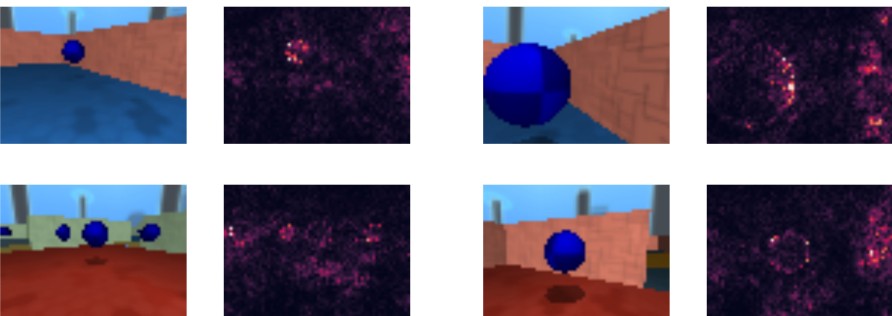

Figure 7: Example agent-view frames and corresponding visuomotor salience visualizations. Note that the ball is more sharply emphasized, suggesting the selected actions were influenced by the affordance of tacking toward the ball.

introduction of redundant clips (see appendix B). We saw benefits in early exploration due to using fragments for more than one time step but lower ultimate performance. Adding more fragments was helpful when those fragments were functionally similar to the standard set and the high-level controller was able to robustly handle those that involved extraneous movements unrelated to loco-motion.

### 3.2 ANALYSIS OF TRAINED HIGH-LEVEL POLICIES

While the query-based approaches did not outperform the discrete control fragment selection (Fig. 6), we include a representative visualization in Appendix A.3 to help clarify why this approach may not have worked well. In the present setting, it appears that the proposal distribution over queries generated by the high-level policy was high variance and did not learn to index the fragments precisely.

On Forage, the high-level controller with discrete selection of control fragments generated structured transitions between fragments (Appendix C). Largely, movements remained within clip or behavior type. The high-level controller ignored some fragments involving transitions from standing to running and left-right turns to use fast-walk-and-turn movements.

To assess the visual features that drove movements, we computed saliency maps (Simonyan et al., 2013) showing the intensity of the gradient of the selected action's log-probability with respect to each pixel: $S_{t;x,y} = \frac{1}{Z} \min(g, \frac{1}{3} \sum_c |\nabla_{I_{x,y,c}} \log \pi(a_t^{HL}|h_t)|)$ with normalization $Z$ and clipping $g$ (Fig. 7). Consistently, action selection was sensitive to the borders of the balls as well as to the walls. The visual features that this analysis identifies correspond roughly to sensorimotor affordances (Gibson, 2014); the agent's perceptual representations were shaped by goals and action.

## 4 DISCUSSION

In this work we explored the problem of learning to reuse motor skills to solve whole body humanoid tasks from egocentric camera observations. We compared a range of approaches for reusing low-level motor skills that were obtained from motion capture data, including variations related to those presented in Liu & Hodgins (2017); Peng et al. (2018). To date, there is limited learning-based work on humanoids in simulation reusing motor skills to solve new tasks, and much of what does exist is in the animation literature. A technical contribution of the present work was to move past hand-designed observation features (as used in Heess et al. (2017); Peng et al. (2018)) towards a more ecological observation setting: using a front-facing camera is more similar to the kinds of observations a real-world, embodied agent would have. We also show that hierarchical motor skill reuse allowed us to solve tasks that we could not with a flat policy. For the walls and go-to-target tasks, learning from scratch was slower and produced less robust behavior. For the forage tasks, learning from scratch failed completely. Finally, the heterogeneous forage is an example of task that integrates memory and perception.

There are some other very clear continuities between what we present here and previous work. For learning low-level tracking policies from motion capture data, we employed a manually specified similarity measure against motion capture reference trajectories, consistent with previous work (Liu et al., 2010; 2015; Peng et al., 2018). Additionally, the low-level policies were time-indexed: they operated over only a certain temporal duration and received time or phase as input. Considerably less research has focused on learning imitation policies either without a pre-specified scoring function or without time-indexing (but see e.g. Merel et al. (2017)). Compared to previous work using control fragments (Liu & Hodgins, 2017), our low-level controllers were built without a sampling-based planner and were parameterized as neural networks rather than linear-feedback policies.

We also want to make clear that the graph-transition and steerable structured low-level control approaches require significant manual curation and design: motion capture clips must be segmented by hand, possibly manipulated by blending/smoothing clips from the end of one clip to the beginning of another. This labor intensive process requires considerable skill as an animator; in some sense this almost treats humanoid control as a computer-aided animation problem, whereas we aim to treat humanoid motor control as an automated and data-driven machine learning problem. We acknowledge that relative to previous work aimed at graphics and animation, our controllers are less graceful. Each approach involving motion capture data can suffer from distinct artifacts, especially without detailed manual editing – the hand-designed controllers have artifacts at transitions due to imprecise kinematic blending but are smooth within a behavior, whereas the control fragments have a lesser but consistent level of jitter throughout due to frequent switching. Methods to automatically (i.e. without human labor) reduce movement artifacts when dealing with large movement repertoires would be interesting to pursue.

Moreover, we wish to emphasize that due to the human-intensive components of training structured low-level controllers, fully objective algorithm comparison with previous work can be somewhat difficult. This will remain an issue so long as human editing is a significant component of the dominant solutions. Here, we focused on building movement behaviors with minimal curation, at scale, that can be recruited to solve tasks. Specifically, we presented two methods that do not require curation and can re-use low-level skills with cold-switching. Additionally, these methods can scale to a large number of different behaviors without further intervention.

We view this work as an important step toward the flexible use of motor skills in an integrated visuomotor agent that is able to cope with tasks that pose simultaneous perceptual, memory, and motor challenges to the agent. Future work will necessarily involve refining the naturalness of the motor skills to enable more general environment interactions and to subserve more complicated, compositional tasks.

### ACKNOWLEDGMENTS

We thank Yuval Tassa for helpful comments. The data used in this project was obtained from mocap.cs.cmu.edu. The database was created with funding from NSF EIA-019621.

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

APPENDICES

## A  ADDITIONAL TRAINING DETAILS

Following Espeholt et al. (2018), all training was done using a distributed actor-learner architecture. Many asynchronous actors interact with the environment to produce trajectories of $(s_t, a_t, r_t, s_{t+1})$ tuples of a fixed rollout length, $N$. In contrast to Espeholt et al. (2018), each trajectory was stored in a replay buffer. The learner sampled trajectories of length $N$ at random and performed updates. Each actor retrieved parameters from the learner at a fixed time interval. The learner ran on a single Pascal 100 or Volta 100 GPU. The plots presented use the steps processed by the learner on the x-axis. This is the number of transition retrieved from the replay buffer, which is equivalent to the number of gradient updates x batch size x rollout length.

We performed all optimization with Adam (Kingma & Ba, 2014) and used hyperparameter sweeps to select learning rates and batch sizes.

| Task | Parameters | | | | |
|------|--------|----------------|-----------|-------|-------------|
|      | unroll | LSTM state size | value MLP | gamma | replay size |
| Go To Target | 10 | 128 | (128, 1) | 0.99 | $10^6$ |
| Walls / Gaps | 20 | 128 | (128, 1) | 0.99 | $10^4$ |
| Forage | 50 | 256 | (200, 200, 1) | 0.995 | $10^4$ |
| Heterogeneous Forage | 200 | 256 | (200, 200, 1) | 0.99 | $10^5$ |

Table 1: Parameters for training the agent on different environments/tasks.

### A.1  SELECTED LOW-LEVEL POLICIES TRAINED FROM MOTION CAPTURE

For the switching controller and control fragments approach we used a standard set of four policies trained from motion capture which imitated stand, run, left and right turn behaviors. In the switching controller, pretrained transitions were created in the reference data. For the control fragments approach, we were able to augment the set without any additional work and the selected policies are described in Table 2.

| Task | Selected policies | Num. control fragments |
|------|-------------------|------------------------|
| Go To Target | stand, run, left turn, right turn | 105 |
| Walls | stand, run, left turn, right turn | 105 |
| Forage | stand, run, left turn, right turn, 2 walk and turns | 183 |
| Heterogeneous Forage | stand, run, left turn, right turn, 2 turns and 2 about-face | 359 |
| Gaps | stand, run, left turn, right turn, 8 jumps | 359 |

Table 2: Selected motion-capture clips for control fragments controller.

### A.2  HETEROGENEOUS FORAGE TASK

In the heterogeneous forage task, the humanoid is spawned in a room with 6 balls, 3 colored red and 3 colored green. Each episode, one color is selected at random and assigned a positive value (+30) or a negative value (-10) and the agent must sample a ball and then only collect the positive ones.

## A.3 HIGH-LEVEL CONTROLLER TRAINING DETAILS

The architecture of the high-level controller consisted of proprioceptive encoder and an optional image encoder which, along with prior reward and action, were passed to an LSTM. This encoding core was shared with both the actor and critic. The details of the encoder are depicted in Fig. A.1.

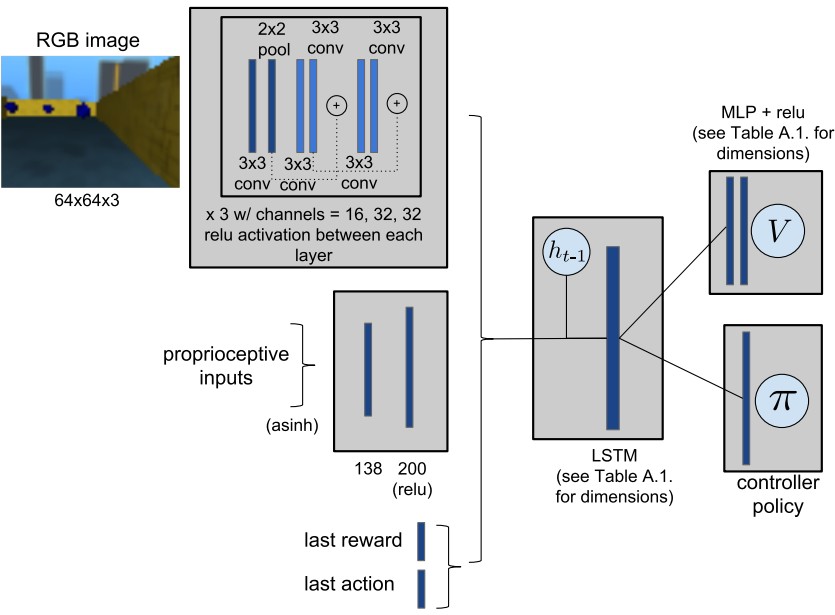

Figure A.1: Complete diagram of high-level agent architecture with encoders.

### A.3.1 STEERING CONTROLLER TRAINING DETAILS

The steering controller policy head took as input the outputs of the LSTM in Fig. A.1. The policy head was an LSTM, with a state size of 128, followed by a linear layer. The linear layer produced the parameters for a 1-D Gaussian. The $\mu$ parameters were constrained by a $\tanh$ and the $\sigma$ parameters were clipped between $[0.1, 1]$.

The policy was trained with a SVG(0) update (Heess et al., 2015). A state-action value / Q function was implemented as an MLP with dimensions in Table 1 and trained with a Retrace target. Target networks were used for Q and updated every 100 training iterations.

The policy was also updated by an additional entropy cost at each time step, which was added to the policy update with a weight of $1e^{-5}$.

### A.3.2 SWITCHING CONTROLLER TRAINING DETAILS

The switching controller policy head took as input the outputs of the LSTM in Fig. A.1. The policy head was an LSTM, with a state size of 128, followed by a linear layer to produce the logits of the multinomial distribution.

The policy was updated with a policy gradient using $N$-step empirical returns with bootstrapping to compute an advantage, where $N$ was equivalent to the rollout length in Table 1. The value-function (trained via V-Trace) was used as a baseline.

The policy was also updated by an additional entropy cost at each time step, which was added to the policy update with a weight of .01.

### A.3.3 DETAILS OF QUERY-BASED ACTION SELECTION APPROACH

We train a policy to produce a continuous feature vector (i.e. the query-action), so the selector is parameterized by a diagonal multivariate Gaussian action model. The semantics of the query-action

will correspond to the features in the control fragment feature-key vectors, which were partial state observations (velocity, orientation, and end-effector relative positions) of the control fragment's nominal start or end pose.

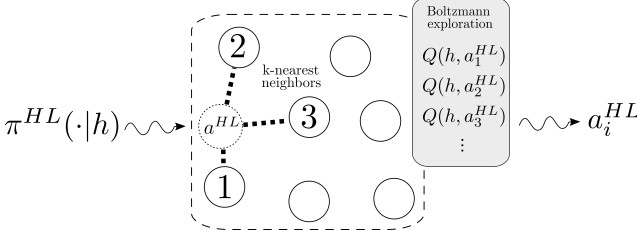

Figure A.2: Illustration of query-based control fragment selection in which a query feature vector is produced, compared with key feature vectors for all control fragments, and the Q-value of selecting each control fragment in the current state is used to determine which control fragment is executed.

In this approach, the Q function was trained with 1 step returns. So, for samples $(s_t, a_t, r_t, s_{t+1})$ from replay:

$$q_{target} = r_t + \gamma \mathbb{E}_{a \sim \pi(\cdot|h_{t+1})}[Q(h_{t+1}, a)]$$

$$\ell_{critic} = \frac{1}{2}||Q(h_t, a_t) - q_{target}||_2^2$$

The total loss is summed across minibatches sampled from replay. Note that for the query-based approach we have query actions which are in the same space as but generally distinct from the reference keys of the control fragments which are selected after the sampling procedure.

The high-level policy emits query actions, which are rectified to reference keys by a nearest lookup (i.e. the selected actions). This leads to two, slightly different high-level actions in the same space. This leads to the question of what is the appropriate action on which to perform both policy updates and value function updates.

To handle this, it proved most stable to compute targets and loss terms for both the query-actions and selected actions for each state from replay. In this way, a single Q function represented the value of query-actions and selected actions. This was technically an approximation as the training of the Q-function pools these two kinds of action input. Finally, the policy update used the SVG(0)-style update (Heess et al., 2015):

$$\ell_{actor} = -\mathbb{E}_{\xi \sim \mathcal{N}(0,1)} Q(h_t, a(h_t, \xi))$$

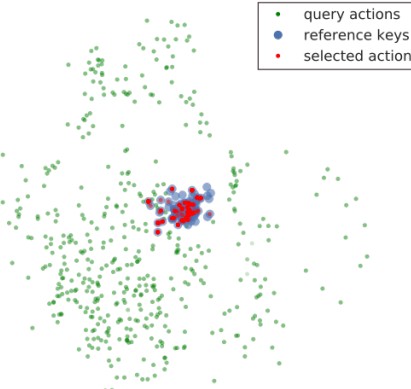

Figure A.3: Visualization (using PCA) of the actions produced by the trained query-based policy (on go-to-target). Query actions are the continuous actions generated by the policy. Reference keys are the feature vectors associated with the control fragments (here, features of the final state of the nominal trajectory of the fragment). Selected actions are the actions produced by the sampling mechanism and they are overlain to emphasize the control reference keys actually selected. The most prominent feature of this visualization is the lack of precision in the query actions. Note that 45/105 control fragments were selected by the trained policy.

We hypothesize that the limited success of this approach is perhaps partly due to the impreciseness in their selectivity (see Fig A.3). After finding these approaches were not working very well, an additional analysis of a trained discrete-selection agent, not shown here, found that the second most

preferred control fragment (in a given state) was not usually the control fragment with the most similar reference key. This implies that the premise of the query-based approach, namely that similar fragments should be preferentially confused/explored may not be as well-justified in this case as we speculated, when we initially conceived of trying it. That analysis notwithstanding, we remain optimistic this approach may end up being more useful than we found it to be here.

## A.4 END-TO-END CONTROLLER TRAINING DETAILS

End-to-end training on each task was performed in a similar fashion to training the low-level controllers. Using the same architecture as in A.1, the policy was trained to output the 56-dimensional action for the position-controlled humanoid. As in the low-level training, the policy was trained with a SVG(0) update (Heess et al., 2015) and Q was trained with a Retrace target. The episode was terminated when the humanoid was in an irrecoverable state and when the head fell below a fixed height.

## B SCALING AND FRAGMENT LENGTH

We compared performance as a function of the number of fragments as a well as the length of fragments. If using a fixed number of behaviors and cutting them into control fragments of various lengths, two features are coupled: the length of the fragments vs. how many fragments there are. One can imagine a trade-off – more fragments might make exploration harder, but shorter temporal commitment to a fragment may ultimately lead to more precise control. To partially decouple the number of fragments from their length, we also compared performance with functionally redundant but larger sets of control fragments.

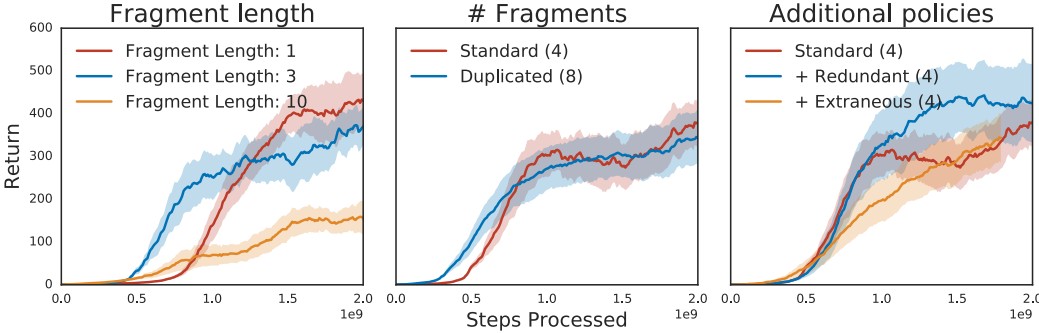

Figure A.4: A. Training was most stable with shorter fragment length (1 / .03 sec or 3 / .09 sec. B. Increasing the number of fragments, by duplicating the original set, did not hurt training performance. C. Additional functionally redundant policies (i.e. 4 vs 8 behaviors, cut into many control fragments) improved training speed, while additional extraneous policies were easily ignored.

Ultimately, it appears that from a strict task-performance perspective, shorter control fragments tend to perform best as they allow greatest responsiveness. That being said, the visual appearance of the behavior tends to be smoother for longer control fragments. Control fragments of length 3 (.09 sec) seemed to trade-off behavioral coherence against performance favorably. Hypothetically, longer control fragments might also shape the action-space and exploration distribution favorably. We see a suggestion of this with longer-fragment curves ascending earlier.

# C  TRANSITION ANALYSES OF AN EXAMPLE POLICY

Figure A.5: Transition density between control fragments for a trained agent on Forage. The background colors reflect density of transitions within a clip/behavior class (darker is denser), and single fragment transition densities are overlain as circles where size indicates the density of that particular transition.

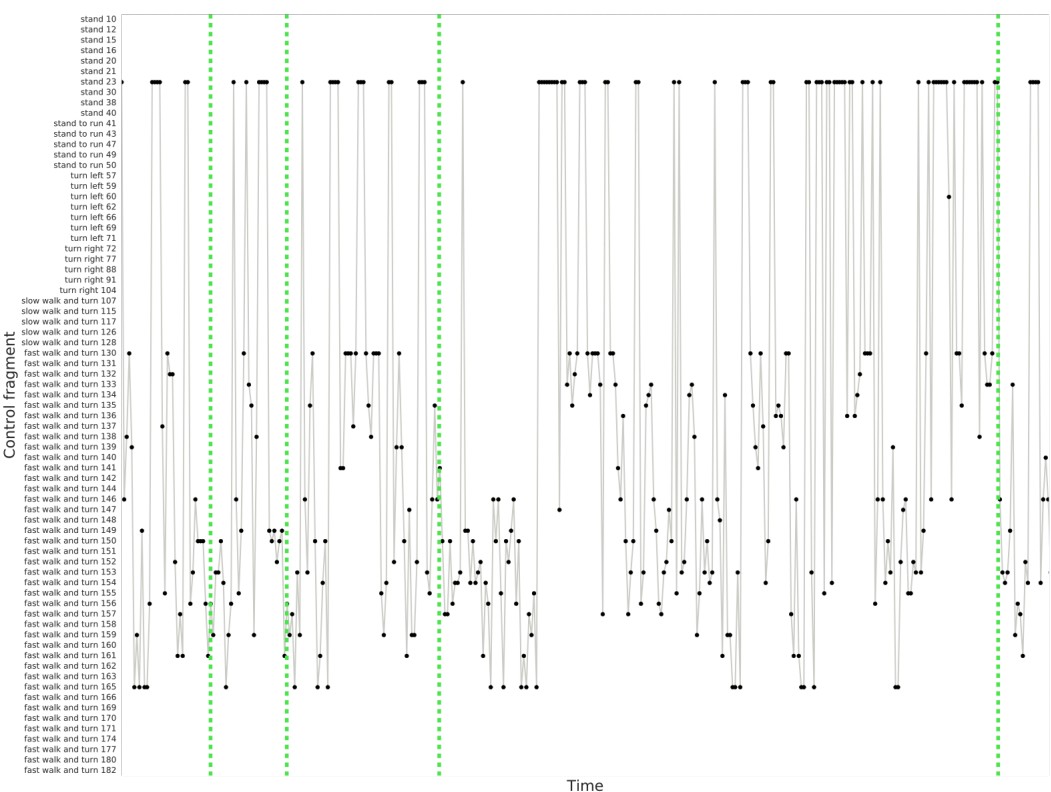

Figure A.6: We depict a timeseries of the behavior of a trained high-level policy on Forage. In this particular trained agent, it frequently transitions to a particular stand fragment which it has learned to rely on. Green vertical lines depict reward acquisition.

