# OpenReview forum: "Hierarchical Visuomotor Control of Humanoids"
_ICLR.cc/2019/Conference_

### Official Review · AnonReviewer3 · 2018-11-02

**Rating:** 6
**Confidence:** 4

**Review:**

1) Summary
This paper proposes a hierarchical reinforcement learning (HRL) method for visual motor control of humanoid agents. The method is decomposed into a high-level controller that takes in visual input and proprioceptive information, and a low-level controller (they compare may ways of doing this) that takes care of the agent’s motor control. In experiments, the proposed method is tested on a variety of RL tasks where the many low-level controllers presented in the paper are compared against each other.

2) Pros:
+ Novel high-level controller that takes in front-view visual information
+ Novel multi-policy low level controller
+ Interesting experimental section

3) Cons:
Numerical comparison to previous methods:
- The only issue I found with this paper is that there is no comparison with other methods. Even if the other methods do not take in front-view visual input, it would be nice to compare with them. Maybe visual inputs results in better high-level controller? Or even show that performance is similar would be an interesting result.

4) Comments:
Jerky transitions in switching controller:
- Due to the fact that one policy takes over after each other based on the high-level controller choice, there is a jerk artifact that shows when the policies are being changed/executed. Did you guys try to add a connection in feature space between policies rather than only passing the state of the agent? This may be able to help with that artifact that sampling noise adds to the actions. Can the authors comment on this?

Steerable controller limited rotation:
- From observing the steerable controller policy in action, it seems the policy learned a steering that is somewhat independent of what the limbs are doing. Maybe adding a mechanism where the leg motion intensity depends more on the direction of movement could be a way to fix the issue where this policy moves to fast for the turning it tries to do. Maybe an energy based objective to minimize the torques or something in that line.

4) Conclusion:
To the best of my knowledge, this paper proposes a novel interesting method for modeling humanoid motor skills with front-view visual input. However, as mentioned above, the paper lacks of numerical comparisons with other methods, and only compares against its own variations which is more of an ablation study. I am willing to increase my review score if the authors successfully address the concerns mentioned above

---

> ### Author Response · Authors · 2018-11-12
> **Response to reviewer comments and questions.**
>
> We thank the reviewer for appreciating the difficulty of the problem and the novelty of the setting, including the use of egocentric vision.
>
> The main concern of this reviewer seems to be about comparison to other methods.  Our core intent with this paper has indeed been a comparison of different methods, adapted from the literature for reusing low-level motor skills.  Previous work (e.g. Merel et al 2017, Peng et al 2018, and others) have attempted to build low-level controllers that incorporate transitions and are conditioned on a pre-specified input parameterization (such as heading direction).  This style of approach is represented in our comparisons by the steering and graph-transitioning approaches.  We also explored the use of control fragments for skill reuse (proposed by Liu et al. 2017) -- we presented results using a default version of this as well as a novel variant.  As far as we are aware, these approaches amount to the current most competitive approaches for motor skill reuse.  Our work goes beyond this by demonstrating that these forms of motor reuse enable visuomotor control and we can, in some cases, be used to solve more complex whole-body tasks than previously done.  Overall, we view this current paper as an integrative work that helps establish and clarify the state of the art involving motor reuse for generic humanoid movement tasks.
>
> Concerning jerky transitions and other visual idiosyncrasies -- we agree that transitions between sub-behaviors are jerky here.  One could attempt to enforce smoothness, however, without adjusting the low level controllers, this would merely amount to a prior or constraint on which transitions would occur and would reduce task performance.  We also agree that there are numerous ways to train the low-level controllers to be a bit smoother -- this is important for computer graphics and would make movements more visually pleasing, but does not necessarily affect task performance.  Importantly, these manual adjustments can require considerable human effort to tune and for this paper we wanted to focus on general techniques for reuse that require as little hand-tuning as possible as we believe this will be critical for scaling to large skill repertoires.

---

### Official Review · AnonReviewer1 · 2018-11-04
**Good paper with great results in interesting environments**

**Rating:** 8
**Confidence:** 3

**Review:**

The paper proposes a control architecture for learning task oriented whole body behaviors  in simulated humanoid robots bootstrapped with motion capture data.

The authors use a hierarchical approach, where the low-level controllers are trained to follow motion captured data whereas the high-level control combines them.
The topic of the paper is interesting and the language is understandable.

The paper discusses and compares different ways to achieve such a higher level control.
It probably won’t be useful for real robots, but will be possibly useful for computer graphics.
I suspect that code will not be published anytime soon, and I am afraid it will be hard to reproduce without. There is a solid software engineering involved and the system has many parameters.

The related work section (or lack thereof) can be improved. What is the advantage of this work over the multi-skill integration in Peng et al 2018? Please explain explicitly in the paper.

The end-to-end approach seems a bit too weak to me. The video shows more artifacts than other similar papers, (cf. Heess et al. 2017). What’s the detail of the training for the end-to-end baseline?

Are the environments randomized in each rollout? If not then this would need an ablation study which ablates memory/vision to prove its claim of integrating vision and memory.
How much is the memory used in the tasks where nothing needs to be memorized?
Is there any noise in the simulations?

One weakness is that the low-level controllers are not adapted any further. That is probably why the fragments outperformed the transition policies etc., because the higher level policy has more flexibility.

Overall, from the perspective of deep learning, I think the paper is novel and provides some insights into different approaches to the problem.

---

> ### Public Comment · (anonymous) · 2018-11-08
> **"I suspect that code will not be published anytime soon, and I am afraid it will be hard to reproduce without."**
>
> It is a DeepMind paper for sure. Saying this is redundant. They never publish any code.

---

> ### Author Response · Authors · 2018-11-12
> **Response to reviewer comments and questions.**
>
> We thank the reviewer for their feedback.
>
> We share your interest in reproducibility. The body we have used is only a slight variation on the DM control suite CMU humanoid (https://github.com/deepmind/dm_control; Tassa et al. 2018); the motion capture data are already available, as is code in the suite to register mocap data to the body. We intend to release the updated body and task environments used in this work. Unfortunately, because of the complexity and inter-linkedness of the agent-level code, along with the requirements for cluster-specific, high-performance compute, we cannot at the present easily release it, but elements of the algorithms are available (e.g., the high-level controllers were trained with the same algorithm as in Espeholt et al., 2018; http://github.com/deepmind/scalable_agent).
>
> Peng et al. 2018 described a few approaches for multi-skill integration:
> The multi-clip reward and skill selector approaches are similar to our conditional tracking along a graph as it involves training a low-level controller based on a few clips.  The details differ slightly, but these approaches are similar in spirit to our baseline of training a low-level controller to switch based on a parametric input.  We will clarify this relationship in the text.
> The composite policy approach in Peng et al. 2018 is more specific and involves selecting subsequent skills based on the relative value function for transitioning to subsequent behaviors -- it is essentially autonomous, so it is less immediately amenable to re-purposing in the context of new tasks and less relevant in the context of the present work.
> Overall though, we emphasize that the Peng et al. 2018 style approaches require manual curation of the mocap data (similar to our transition-graph and steerable tracking approaches), and we aimed with this work as well to explore approaches (such as control fragments) that require very little/no manual curation.
>
> The comparison with Heess et al. 2017 is a good question.  Sensible looking locomotion behavior can be achieved in a variety of ways (environmental constraints, appropriate shaping rewards, task setup including multiple tasks or curricula).  Heess et al. 2017 uses a simpler reward that encourages a constant forward velocity together with environmental variations, on a simpler humanoid model. In our own end-to-end experiments, especially slow movements and transitions between standing and moving have proven relatively difficult. The comparison here is mostly meant to demonstrate that the particular go-to-target task setup does not easily give rise to naturalistic looking behavior when trained end-to-end. However, we can expand on the details of the end-to-end training in the text.
>
> In each episode of a task, the environment is randomized. In the Walls and Gaps task the layout of the track is randomized, requiring the agent to use vision to navigate. In the Forage task, the maze layout and placement of rewards is randomized. Finally, in the heterogeneous forage task, each orb color is assigned a random positive or negative reward. In this case the agent must match color to reward in the episode to only eat the “good” orbs. We can clarify in the text that not all tasks require memory.  More broadly, the use of memory is a little subtle insofar as the high-level agent sees information at each low-level timestep, but only acts to switch among the low-level controllers (for control fragments, e.g. every 3 timesteps; it at least makes sense that some state information seen by the high-level controller between selection timesteps could inform subsequent selection, but we agree we don’t assess this explicitly).
>
> We agree that adapting low-level skills is important and this is a clear direction for future research.

---

### Official Review · AnonReviewer2 · 2018-11-12

**Rating:** 5
**Confidence:** 3

**Review:**

1) Summary
The authors propose an interesting hierarchical reinforcement learning method that makes use of visual inputs as well as proprioception for locomotion of humanoid agents. The low-level controllers make use of “motion capture” data and are expected to form a set of movement primitives that can be used by a higher-level controller that has vision and memory. Their method is tested on a variety of tasks and different choices of low-level controller are explored.

2) Pros
+ Combining vision, memory, and motor control
+ Allows the high-level controller to operate at a coarser time scale
+ The set of low-level movement primitives can be extended by using more mocap data

3) Cons
- No comparison to earlier work
- Highly unnatural motions even though it makes use of mocap data
- Sample inefficient: more than 1 billion time-steps to train the high-level controller

4) Comments
Showing that the agent can provide suitable solutions for these tasks using raw vision input is indeed interesting, however it is not clear what the main contribution of the paper is as the authors fail to compare their results with earlier work. It would be useful if the authors could cover the related work in more depth in order to motivate their method and contrast it with the existing solutions. As an example, DeepLoco (Peng et al. 2017) solves a similar problem in which they use an egocentric heightmap instead of direct visual input, hence a formal consideration of the trade-offs would be informative.

In addition, the appeal of using hierarchical reinforcement learning is to divide up the task into easier chunks that can be solved easier, however it is not obvious how well this method succeeds at this task, keeping in mind that the high-level controller takes in the order of 1 billion time-steps to learn most tasks (5 billion in the case of “Heterogeneous Forage”).

In the end, an ablation study could be useful since the authors make plenty of novel design decision, yet their effect on the final performance is not clear.


6) Questions
- Is is possible to entirely remove proprioception from the input to the high-level controller or at least use just a small portion of it? How do the results compare in this case?

- How robust is “cold-switching” between control fragments? Is it possible to transition between most fragments without losing balance or does the high-level controller have to be extremely careful as to which combination it should use? The former case would suggest that this method is indeed useful as a hierarchical method. However the latter case might imply that the hierarchical method is failing and the higher level controller’s task has not been made much easier than the original problem itself.

- Table 2 describes the mocap clips used to train the low-level controllers in each task. What is the effect of choosing different sets of motions? Specifically how well does the steerable controller work if walking motions were used for the “Go To Target” and “Walls” tasks rather than running motions? Presumably, this can result in a more flexible controller which allows sharper turns without loosing balance.

- The network in Figure A.1 gets the last action as an input. Why is this required? Especially since the LSTM unit can learn to remember any information related to the previous actions.

- How does the supervised pre-training described in section 2.1 effect the training of low-level controllers? Is it used as a speed-up mechanism or a way of escaping local minima?

- In section 2.1 the authors mention that the episodes are “terminated when the pose deviates too far from the trajectory”. I believe this termination criteria was not present in the earlier works (Peng et al. 2018), then what is the effect of adding such a criterion? Can this make the learned agent less robust as it will not learn to recover from larger perturbations?


7) Conclusion
The method and the results are interesting but further comparison with existing work is required.

---

> ### Author Response · Authors · 2018-11-14
> **Response to reviewer comments and questions (part 1/2).**
>
> We thank the reviewer for a careful reading and for these questions. We will first address the three “cons” described above.
>
> Comparison to earlier work: We have provided a thorough set of comparisons and investigation of the components of the system where possible. In particular, our low-level controllers are built using a variety of techniques, encompassing the techniques in Liu et al., 2017, and Peng et al., 2018. Therefore, at the low-level policy level, we are indeed primarily benchmarking and scaling up existing techniques. At the composite system level, however, we feel that the question is ill-posed. Consider, for example, the DeepLoco paper (Peng et al., 2017): the motion capture data used in this work were manually selected and preprocessed. While undoubtedly excellent work, this step of manual curation makes it difficult to understand what an objective algorithm comparison would mean. Our work aimed to minimize this manual curation. As we demonstrated, the control fragments approach scaled well with the inclusion of redundant or irrelevant motion capture data. We examined the application of existing techniques requiring little manual curation to the problem of hierarchical, memory-based visuomotor control of humanoids.
>
> Jerkiness: The movements were admittedly slightly jerky in the control fragment model due to switching among the fragments. In the appendix, we also demonstrate the trade-off between longer fragments, which would result in smoother motion, and task performance.  For the steering and graph switching approaches, any jerkiness was merely a consequence of the artistry with which mocap is curated: we generally preferred methods that enable scaling to large numbers of clips and the solution of new high-level tasks, instead of manual curation of motion capture data.
>
> Sample efficiency: While we agree that 1 billion timesteps seems large, we would emphasize that this is standard at present (e.g. Peng et al. 2018 uses within 1 order of magnitude of that number for learning skill-selection behavior even without vision-based control).  Additionally, for each higher-level task, we provided a comparison with simple rolling-ball body. This body was used to demonstrate the difficulty of the task independent of the control problem.
>
> We would like to emphasize that there is limited learning-based work on humanoids in simulation reusing motor skills to solve new tasks, and this work is novel in this regard: certainly, the use of an egocentric camera to guide visuomotor behavior of humanoids is little studied.  Other work on simulated humanoid control has primarily used state features designed to provide input to the agent about terrain (Peng et al. 2017, Heess et al. 2017, Peng et al. 2018). A contribution of the present work was to move past hand-designed features towards a more ecological observation setting.
>
> A scientific contribution of this work was also to show that hierarchical motor skill reuse enabled tasks that were unsolvable with flat policy learning to be solved. We clearly demonstrated this. For the walls and go-to-target tasks, learning from scratch was slower and produced less robust behavior; for the forage tasks, learning from scratch failed completely. Since control fragments were the most compelling LL control approach among those we considered, we did a thorough study of the effect of fragment length, number of fragments, as well as introduction of redundant clips (see appendix B).

---

> > ### Author Response · Authors · 2018-11-14
> > **Response to reviewer comments and questions (part 2/2)**
> >
> > Replies to specific questions:
> > 1) We have not considered the case where high level controllers are information limited -- rather, we view the more interesting asymmetry being that the low level controller only has proprioception.  Depending on the task, it seems likely that only providing vision to the high level controller may do as well as also providing proprioception.
> >
> > 2) Switching among control fragments cannot really be assessed for only a single transition as it might only become clear that a switch from fragment A → B was a bad choice after realizing that from the state arrived upon due to the sequence of actions (select A, select B), there are no good subsequent options.  As such, the appropriate way to examine how flexibly it is possible to switch among control fragments is to examine transition behavior of trained policies.  We depicted an example of this in Figure A.5 (for the go-to-target task).  We see some diversity of transitions, especially within the fast walk and turn clip, which makes sense for this task.
> >
> > 3) The graph transition and steerable approaches require significant manual curation -- mocap clips must be segmented by hand, possibly manipulated by blending/smoothing clips from the end of one clip to the beginning of another.  This process takes human labor and to do it well requires considerable skill as an animator.  As researchers with a machine learning orientation, it seems implausible to us that the most productive path forward for motor control is to hand-curate and animate specific behavioral transition, but indeed we have made a sincere attempt to implement some of these baselines.  Through this work, we have found that we can much more rapidly develop methods that scale to more complex tasks if we avoid hand-designing the reuse of low level skills and instead use methods that require little to no human curation.  This message we view as a strong takeaway for ourselves, and we wanted to communicate this to the readers of the paper.
> >
> > 4) Providing the previous action to as an observation to the policy is a minor design choice that is not critical to this approach.  We followed similar agent architectures as in Mnih et al. 2016, Espeholt et al. 2018.
> >
> > 5) The supervised pretraining for the mocap tracking controllers does both help avoid local optima for the RL training and speed up the training.  We can clarify this in the text.
> >
> > 6) It is relatively common for early termination to be used.  Heess et al. 2017 and Peng et al. 2018 have both used early terminations, and early terminations can generally be based on contact of body parts with the ground.  Often papers are slightly unclear about what termination criterion they use so we stated that previous work “ terminated when the pose deviates too far from the trajectory or when the body falls”.  The subsequent sentence clarifies that “Our specific termination condition triggers if parts of the body other than hands or feet make contact with the ground.”  We will edit this section further for clarity.

---

> > > ### Comment · AnonReviewer2 · 2018-11-20
> > > **Final Review Comments**
> > >
> > > The authors claim that this method improves upon the earlier work by substantially decreasing the amount of manual curation needed, however I still cannot see any real difference in the level of manual work required. This method as well as the earlier work (Peng et al. 2017 and Peng et al 2018) use existing pre-cleaned mocap data. Though I understand that importing the data to be usable in the RL framework is burdensome, I still believe that this step is shared between the approach in this work as well as the earlier works. Therefore, although I recognize the value in the authors' goal of using as little manual curation as possible, I don't believe this paper takes a substantial step towards this goal.
> > >
> > > In regards to the hierarchical structure that was presented, I don't see much in terms of novelty in this framework and I am not convinced that this method is effective enough in making the task much easier for the higher-level controller.

---

### Public Comment · (anonymous) · 2018-10-18
**Strange humanoid model**

Hi,

What was a reasoning behind the creation and using such a strange human model? It looks and behaves very unrealistically, for example, hands are much smaller than a real human with the same height should have.

Actuators look too weak - humanoid can't jump and run well enough, running looks very heavy more like a usual walking.

What was a motivation for such a design? This humanoid model has much more degrees of freedom and looks like it was supposed to be more realistic and closer to the real human, compared to the traditional 23 DoF but it's not the case with its wrong proportions and motor strengths.

---

> ### Author Response · Authors · 2018-10-22
> **Reply to anonymous comment about humanoid model**
>
> This body was created in other work and is publicly available -- see the DM control suite github (https://github.com/deepmind/dm_control; which has two branches with variants of the “Humanoid_CMU”). As stated in the DM control suite write-up, the Humanoid_CMU segment lengths and pose parameterization are based on a subject in the CMU mocap database, so the body is easy to set to poses obtained from that database.
>
> Humans vary quite significantly in actuation strength. The actuation strengths of the model can be seen in the DM control suite model. These numbers appear relatively strong for a body of this mass, and in other experiments it is indeed capable of dynamic, humanlike movements including running and jumping and acrobatics.  The present work does not emphasize or require highly dynamic movements; rather, we study schemes for reusing basic motor skills for solving high-level tasks with minimal manual curation.

---

> > ### Public Comment · (anonymous) · 2018-11-14
> > **Still doesn't reply to the main question**
> >
> > Thanks for the reply! But it seems like some points still have to be clarified. I know that that this Humanoid_CMU model is available in Deepmind Control Suite. But this fact doesn't shed any light on some of the details of the design choice:
> >
> > 1) Why hands are much smaller than for a human of the same height and feet are bigger?
> >
> > 2) How robust is your method? Is it highly sensitive to the parameters above and will work only for these proportions and with feet larger than real human ones?

---

### Author Response · Authors · 2018-11-15
**Updated submission in response to reviewer feedback.**

Taking all of the reviewer comments into consideration, the first round of exchange has prompted us to revise some of how we communicated the ideas.  Specifically, in addition to localized updates in direct response to reviewer comments, the substantial changes are that we revised the intro paragraphs to section 2.2, now titled “Varieties of low-level motor control”, and we substantially changed the discussion (section 4).  We hope these revisions make clearer how what we have done relates to existing approaches.  We again thank the reviewers for their input.

---

### Comment · Area_Chair1 · 2018-11-19
**further reviewer thoughts -- responses to the author's replies?**

Thanks to everyone for the detailed reviews, and the authors for their detailed replies.

Reviewers:  please advise as to whether the replies have influenced your evaluation and your score for the paper.
Your input is greatly appreciated.

Note that there is a convenient way to see the revision differences: select "Show Revisions" on the review page, and then select the check-boxes for the two versions you wish to compare.

-- area chair

---

### Meta-Review · Area_Chair1 · 2018-12-16
**accept;  vision-enabled/memory-enabled/mocap-mimicing humanoid**

**Confidence:** 4
**Recommendation:** Accept (Poster)

**Metareview:**

A hierarchical method is presented for developing humanoid motion control,
using low-level control fragments, egocentric visual input, recurrent high-level control.
It is likely the first demonstration of 3D humanoids learning to do memory-enabled tasks using only
proprioceptive and head-based ego-centric vision. The use of control fragments as opposed
to mocapclip-based skills allows for finer-grained repurposing of pieces of motion, while
still allowing for mocap-based learning

Weaknesses: It is largely a mashup up of previously known results (R2).  Caveat: this can be said for all research
at some sufficient level of abstraction. The motions are jerky when transitions happen between control fragments (R2,R3).
There are some concerns as to whether the method compares against other methods; the authors note
that they are either not directly comparable, i.e., solving a different problem, or are implicitly
contained in some of the comparisons that are performed in the paper.

Overall, the reviewers and AC are in broad agreement regarding the strengths and weaknesses of the paper.

The AC believes that the work will be of broad interest. Demonstrating memory-enabled, vision-driven,
mocap-imitating skills is a broad step forward. The paper also provides a further datapoint as
to which combinations of method work well, and some of the specific features required to make them work.

The paper could acknowledge motion quality artifacts, as noted by the reviewers and
in the online discussion.  Suggest to include  [Peng et al 2017] as some of the most relevant related HRL humanoid control work, as per the reviews & discussion.